# Knowledge, attitude, and practice of the rural community about cutaneous leishmaniasis in Wolaita zone, southern Ethiopia

**Bereket Alemayehu**[1,2]*, **Abraham Getachew Kelbore**[3], **Mihiretu Alemayehu**[4], **Chimdesa Adugna**[1], **Tessema Bibo**[1], **Aberham Megaze**[1], **Herwig Leirs**[2]

**1** Department of Biology, College of Natural and Computational Sciences, Wolaita Sodo University, Wolaita Sodo, Ethiopia, **2** Evolutionary Ecology Group, University of Antwerp, Antwerp, Belgium, **3** Department of Dermatology, College of Health Sciences and Medicine, Wolaita Sodo University, Wolaita Sodo, Ethiopia, **4** School of Public Health, College of Health Sciences and Medicine, Wolaita Sodo University, Wolaita Sodo, Ethiopia

* bereketalemayehu@gmail.com

**Data Availability Statement:** All relevant data are within the paper and its Supporting information files.

## Abstract

### Background

Cutaneous leishmaniasis (CL) is a neglected tropical disease that is caused by a *Leishmania* parasite and transmitted by the bite of infected female sandflies. Community awareness is an essential component of disease control and prevention. Therefore, this study aimed to assess the community's knowledge, attitude, and practice toward CL in Wolaita zone, southern Ethiopia.

### Methods

A community-based cross-sectional study design was employed to include 422 study subjects selected using a systematic sampling technique from two districts, Kindo Didaye and Sodo Zuria. A pretested structured questionnaire was used to collect data from the household heads. Bivariate and multivariate logistic regression analyses were performed to determine the relationship between the participants' knowledge about CL and sociodemographic characteristics.

### Results

Out of the 422 study participants, only 19% had good knowledge of CL in general. Most (67.1%) of the respondents knew CL by its local name ("bolbo" or "moora") though this knowledge varied highly over the study districts. The majority (86.3%) of respondents did not know how CL is acquired, though they considered CL a health problem. Most (62.8%) respondents believed that CL was an untreatable disease. Most (77%) participants responded that CL patients preferred to go to traditional healers for treatment. Herbal treatment was the most (50.2%) used to treat CL. Knowledge about CL was significantly associated with sex, age, and study districts.

**Funding:** This work was funded by the Flemish Interuniversity Council (VLIR-UOS, ET2019TEA485A102). The funders had no role in study design, data collection, and analysis, decision to publish, or preparation of the manuscript.

**Competing interests:** The authors have declared that no competing interests exist.

**Abbreviations:** CL, Cutaneous Leishmaniasis; GPS, Global Positioning System; KAP, Knowledge, Attitude, and Practice; NTD, Neglected Tropical Disease; WHO, World Health Organization.

## Conclusion

The overall knowledge, attitude, and practice about CL and its prevention in the study area were low. This emphasizes the need to implement health education and awareness campaign to reduce the risk of CL infection. Policymakers and stakeholders should also give due attention to the prevention and treatment of CL in the study area.

## Introduction

Cutaneous leishmaniasis (CL) is a neglected tropical disease (NTD) and is caused by a protozoan parasite of the genus *Leishmania*. It is a vector-borne disease transmitted by the bite of infected female sandflies of various species [1]. The disease is the most common form of leishmaniasis, which causes skin lesions on the exposed parts of the body, leaving life-long scars and serious disability or stigma. WHO estimated that 600,000 to 1 million new cases of CL occur worldwide annually, though this might be an underestimation due to misdiagnosis and lack of reporting. About 95% of CL cases occurred in the Americas, the Mediterranean Basin, the Middle East, Central Asia, and the Sub-Saharan region. In 2018, of the 200 countries and territories reported to WHO to have leishmaniasis, 44% were considered endemic for CL [2]. In endemic countries, the transmission cycle of CL is very complex, involving various species of parasites, sandfly vectors, and reservoir hosts [3, 4].

Ethiopia is one of the world's CL-endemic countries, with an estimated incidence of 20,000 to 50,000 cases annually [5, 6]. Almost all CL infection in Ethiopia is caused by a *L. aethiopica* parasite. Infection with this parasite can result in a mild, localized CL, to a more severe form, diffused CL [7–9]. The Ethiopian CL is reported to be zoonotic, involving hyraxes as reservoir hosts, and the transmission to the human host is effected through blood-feeding *Phlebotomine* sandflies [3, 8–10].

Studies have reported the magnitude of CL and its ecology in Ethiopia until recently [3, 11–14]. However, the disease continues to be a public health problem in the country. There is a lack of knowledge about CL infection among communities in the country, which hinders community participation in disease prevention and control [15, 16]. The national master plan of NTDs of Ethiopia has prioritized community engagement as one of the core elements to achieve the national goals of preventing and controlling NTDs, including CL [17]. Identifying gaps in knowledge, attitude, and practices can guide the development of locally adapted community interventions.

Rural communities bear the highest disease burden [18–20]. The lack of health facilities and inaccessibility of appropriate treatments worsen the situation [21]. Due to a lack of awareness, there are also misconceptions about CL among remote communities [22]. Therefore, community-based knowledge, attitude, and practice (KAP) studies are essential to assess such awareness and perceptions and make the information available mainly for health sectors. However, only a few community-based KAP studies towards CL are available in Ethiopia [18, 23, 24], and none existed for Wolaita zone, southern Ethiopia. Therefore, the present study aimed to assess the knowledge, attitude, and practice about CL in that area.

## Materials and methods

### Study area

The study was conducted in Wolaita zone, southern Ethiopia (Fig 1). The zone is situated at a distance of 330 Km away from Addis Ababa, the Ethiopian capital. The administrative areas in

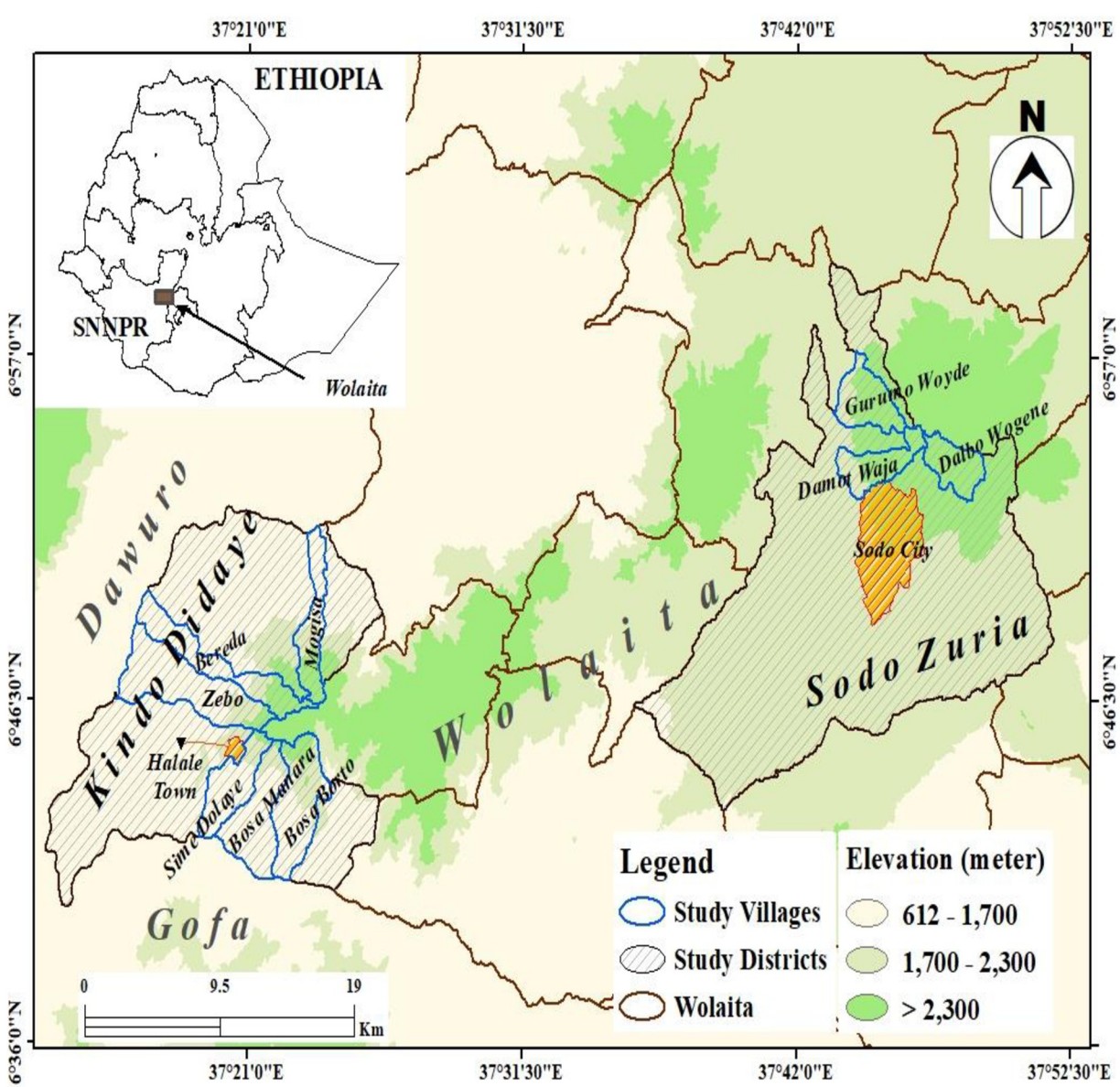

**Fig 1. A location map of the study area (created with ESRI ArcGIS Desktop 10.8).**

Wolaita zone are structured in *woredas* (districts) and *kebeles* (villages). There are seventeen rural districts and three town administrations in the zone. Wolaita zone is one of Ethiopia's most densely populated zones, with over 342 persons living per square kilometer [25]. The majority of the population in the zone resides in rural districts where the primary livelihood is agriculture [26]. According to the zonal reports, although the existing health facilities are insufficient to address the service needs of the large population in the zone, there are significant recent improvements regarding health care provision and infrastructure development. However, traditional treatment is still largely practiced for some diseases, mainly among rural communities.

## Study design

A community-based cross-sectional study was conducted from August to October 2020 in two selected districts of Wolaita zone, Kindo Didaye and Sodo Zuria, based on CL presence

information obtained from the local and the zonal health offices. Information about CL presence in the districts was also obtained from the farmer community during reconnaissance surveys conducted in the zone's mid-highland areas. By using the Garmin 72H GPS apparatus, villages having areas above an altitude of 1700 m.a.s.l. were taken: six villages from Kindo Didaye district (Bosa Borto, Bosa Manara, Sime Dolaye, Zebo, Bereda, and Mogisa) and three villages from Sodo Zuria district (Gurumo Woide, Damot Waja, and Dalbo Wogene) were included in this study. The purposive focus on the study areas with the specified altitude was because CL is reported from highland or mid-highland areas in Ethiopia [3, 8, 10, 27].

## Sample size and sampling

The sample size was calculated using the single population proportion statistical formula for health studies [28], $n = Z^2 P(1-P)/d^2$, with the following assumption: n = the number of study subjects (household heads), **Z** is a critical value (1.96) at 95% confidence level, P = an anticipated population proportion assumed to be 50% to obtain a maximum sample size since there was no previously established report on the KAP about CL for the study area, d = the level of precision or margin of error (5%). Therefore, the minimum sample size of the household heads required for this study was 384. After adding a 10% non-response rate, the total sample size was calculated to be 422. To allocate the sample size for each study district (village in the district), the calculated sample size was proportionally divided by the total households present in the CL suspected areas. Every fifth household was systematically selected, and the household head was approached to gather information about the KAP regarding cutaneous leishmaniasis and its prevention.

## Data collection procedure

A total of thirty-six data collectors and nine supervisors collected the data. The two-day training was given to data collectors and supervisors before the data collection. Pretesting was done in a neighboring village (not included in the actual data collection) in 5% of the sample size. The data were collected using a pretested structured questionnaire (S1 File). The questionnaire contained four parts: part one related to the socio-demography of study participants, part two on knowledge regarding CL, part three on practice related to CL prevention and treatment, and part four on attitude towards CL. The questionnaire was prepared in English and then translated into Amharic or the local language, Wolaittatto, as appropriate and back-translated into English to check for consistency.

## Scoring knowledge

Participants' knowledge was scored according to the method described in other studies [15, 29, 30] with some modifications to adapt to the present study. For each question, a score of 1 point was assigned for answers that were considered to categorize a respondent as knowledgeable, and a score of 0 point was assigned for answers that were considered to categorize a respondent not knowledgeable. The knowledge was assessed using a composite variable created from 5-item questions that assessed the knowledge of participants towards CL. For the knowledge questions, the answers considered to categorize a respondent knowledgeable were: 1) "yes" for "do you know CL?"; 2) "yes" for "do you know zoonosis?"; 3) "by insect bite" for "do you know CL mode of transmission?"; 4) "yes" for "do you know sandflies?" and 5) all except "I don't know" for "do you know the sign and symptoms of CL?". The total knowledge scores ranged from 0 to 5. The knowledge scores between 0 and 2 were considered to indicate poor knowledge, while scores between 3 and 5 were considered to indicate good knowledge.

## Data analysis

Data analysis was performed with R version 4.1 after entering, cleaning, and organizing the data in Epidata version 4.6.0.2. Descriptive statistics (frequency, percentage and mean) were used to describe the socio-demographic variables. Categorical variables were presented using frequencies and percentages. Bivariate and multivariate logistic regression analyses were performed to determine the relationship between the participants' knowledge about CL and socio-demographic characteristics. Possible associations were measured using an adjusted odds ratio (AOR) with 95% CI, and a p-value of less than 0.05 was considered statistically significant.

## Ethics statement

The study was reviewed and approved by the Institutional Review Board of Wolaita Sodo University under the reference number WSU41/12/1225. Subsequently, Permissions were obtained from Wolaita Zone Health Bureau and the respective District authorities. Written informed consent was obtained from each study participant. All demographic data of the study participants were kept confidential and anonymized before analysis.

# Results

## Socio-demographic profiles of the participants

Of the 422 household heads who participated in this study, 367 (87%) were males, and 55 (13%) were females. The average age of respondents was 43.52 years, with a standard deviation of 10.38 years. The majority (226, 53.6%) of the participants were aged less than or equal to 40 years. Most (368, 87.2%) of the participants were farmers, while the rest were non-farmers. With regard to education, 288 (68.2%) of the participants hadn't had a formal education. The mean family size of the households was 5.51 (SD1.761, range 1–10), with ≤5 individuals living per household in 221 (52.4%) houses (Table 1).

**Table 1. Socio-demographic profiles of the household heads.**

| Variables | Frequency | Percent |
|---|---|---|
| Sex | | |
| Male | 367 | 87.0 |
| Female | 55 | 13.0 |
| Age group (year) | | |
| ≤40 | 226 | 53.6 |
| >40 | 196 | 46.4 |
| Occupation | | |
| Farmer | 368 | 87.2 |
| Non-farmers | 54 | 12.8 |
| Status of education | | |
| No formal education* | 288 | 68.2 |
| Formal education | 134 | 31.8 |
| Family size | | |
| ≤5 | 221 | 52.4 |
| >5 | 201 | 47.6 |

*Refers to no formal school attended.

## Knowledge of the participants about CL

CL in the area has local names, "Bolbo" and "Moora," both meaning "disfigure." Most (283, 67.1%) respondents knew CL by its local name though this knowledge varied over the study districts. The majority (369, 87.4%) of respondents hadn't known about zoonosis in general. Though 58 (13.7%) of study participants responded to insect biting as a mode of CL transmission, none knew sandflies. Concerning signs and symptoms of CL, the participants responded that CL disfigures the skin (24.6%), causes lasting wounds (19.7%), mainly occurs on the face (17.5%), and causes pain (11.4%). Overall, only 80 (19%) participants had good knowledge about CL (Table 2).

## The attitude of the participants toward CL

One hundred fifty-seven (37.2%) participants thought CL is treatable. However, most (318, 75.4%) believed that CL patients should not receive modern medication to treat CL. The majority (302, 71.6%) thought CL was a health problem in the area. A significant number (185, 43.8%) of the respondents had bad feelings about meeting CL patients. Most (70.9%) respondents believed CL was not a stigmatizing disease (Table 3).

## Practice toward CL prevention and treatment

Three hundred twenty-five (77%) of the study participants responded that CL patients preferred to go to the traditional healers seeking treatment. Herbal treatment was the most used (212, 50.2%), followed by burning (100, 25.1%) to treat CL. There was no CL control at a

**Table 2. Knowledge of the participants about cutaneous leishmaniasis in Wolaita zone, south Ethiopia.**

| Variables | Frequency | % |
|---|---|---|
| Know CL | | |
| Yes | 283 | 67.1 |
| No | 139 | 32.9 |
| Know zoonosis | | |
| Yes | 53 | 12.6 |
| No | 369 | 87.4 |
| Know CL mode of transmission | | |
| Contact with an infected person | 140 | 33.2 |
| Contact with infected animals | 5 | 1.2 |
| By insect bite | 58 | 13.7 |
| I don't know | 219 | 51.9 |
| Know sandfly | | |
| Yes | 0 | 0 |
| No | 422 | 100 |
| Know the signs and symptoms of CL | | |
| Mostly occur on the face | 74 | 17.5 |
| Disfigure the skin | 104 | 24.6 |
| Cause lasting wound | 83 | 19.7 |
| Pain | 48 | 11.4 |
| I don't know | 113 | 26.8 |
| Knowledge (overall) | | |
| Good | 80 | 19 |
| Poor | 342 | 81 |

**Table 3. The attitude of the respondents about CL in Wolaita zone, south Ethiopia.**

| Variables | Frequency | % |
|---|---|---|
| CL is treatable | | |
| Yes | 157 | 37.2 |
| No | 265 | 62.8 |
| CL patients should receive modern medication | | |
| Yes | 104 | 24.6 |
| No | 318 | 75.4 |
| CL is a health problem in the area | | |
| Yes | 302 | 71.6 |
| No | 120 | 28.4 |
| CL is a stigmatizing disease | | |
| Yes | 123 | 29.1 |
| No | 299 | 70.9 |
| I feel bad when meeting CL patients | | |
| Yes | 185 | 43.8 |
| No | 237 | 56.2 |

community level. Only 23 (5.5%) respondents used bed nets. The indoor residual spray was also practiced by only 14 (3.3%) of the respondents (Table 4).

## Factors associated with knowledge about CL

Both bivariate and multivariate logistic regression analyses showed that age, sex, and study districts were significantly associated with the overall knowledge of the respondents about CL (P<0.05). However, family size, occupation, and educational status were not associated with the respondents' overall knowledge. There were increased odds of having good knowledge among participants who were males (AOR = 4.162, 95% CI = 1.386, 12.499), aged 40 years or

**Table 4. Practice toward CL treatment and prevention in Wolaita zone, south Ethiopia.**

| Variables | Frequency | % |
|---|---|---|
| Treatment preference | | |
| Traditional healers | 325 | 77 |
| Hospitals/clinics | 97 | 23 |
| Type of medication CL patients receive | | |
| Herbal medicine | 212 | 50.2 |
| Burning | 106 | 25.1 |
| Modern medicine | 104 | 24.6 |
| Community-based CL control | | |
| Yes | 0 | 0 |
| No | 422 | 100 |
| Use bed net | | |
| Yes | 23 | 5.5 |
| No | 399 | 94.5 |
| Use insecticides indoor | | |
| Yes | 14 | 3.3 |
| No | 408 | 96.7 |

**Table 5. Association of knowledge about CL with the socio-demographic profiles of the respondents in Wolaita zone, south Ethiopia.**

| Variables | Knowledge about CL | | COR (95% CI) | AOR (95% CI) | P value for AOR |
|---|---|---|---|---|---|
| | Good n (%) | Poor n (%) | | | |
| Sex | | | | | |
| Male | 76 (20.7) | 291 (79.3) | 3.330 (1.167, 9.502)* | 4.162 (1.386, 12.499) | 0.011 |
| Female | 4 (7.3) | 51 (92.7) | 1 | 1 | |
| Age group (year) | | | | | |
| ≤40 | 58 (25.7) | 168 (74.3) | 2.731 (1.600, 4.660)* | 2.020 (1.121, 3.639) | 0.019 |
| >40 | 22 (11.2) | 174 (88.8) | 1 | 1 | |
| Family size | | | | | |
| ≤5 | 35 (15.8) | 186 (84.2) | 0.652 (0.400, 1.065) | 0.658 (0.385, 1.123) | 0.125 |
| >5 | 45 (22.4) | 156 (77.6) | 1 | 1 | |
| Occupation | | | | | |
| Non-farmer | 15 (27.8) | 39 (72.2) | 1.793 (0.933, 3.445) | 1.233 (0.549, 2.769) | 0.612 |
| Farmer | 65 (17.7) | 303 (82.3) | 1 | 1 | |
| Education | | | | | |
| Formal | 32 (23.9) | 102 (76.1) | 1.569 (0.948, 2.596) | 1.728 (0.928, 3.217) | 0.085 |
| Non-formal | 48 (16.7) | 240 (83.3) | 1 | 1 | |
| Study district | | | | | |
| Kindo Didaye | 75 (27.7) | 196 (72.3) | 11.173 (4.407, 8.327)* | 12.379 (4.737, 32.346) | 0.000 |
| Sodo Zuria | 5 (3.3) | 146 (96.7) | 1 | 1 | |

Variables adjusted: Sex, Age group, Family size, Occupation, Education, and Village

*Significant association (P ≤ 0.05) for COR

COR: crude odds ratio, AOR: adjusted odds ratio, CI confidence interval.

below (AOR = 2.020, 95% CI = 1.121, 3.639), and those who lived in Kindo Didaye district (AOR = 12.379, 95% CI = 4.737, 32.346) than their counterparts (Table 5).

## Discussion

The present study assessed the knowledge, attitude, and practice (KAP) of the rural community about CL in Kindo Didaye and SodoZuria districts of Wolaita zone, southern Ethiopia. Although CL was perceived as a serious health problem in the study community, our findings showed poor knowledge about CL and little use of disease prevention strategies.

The present study participants knew CL by its local name, either "bolbo" or "moora," meaning disfigurement of the exposed parts of the body. Almost all the community groups in Kindo Didaye district knew CL in the local name, while in Sodo Zuria district, none had the knowledge. The reasons for such differences in the usage of CL local name in the present area could be due to the extent of disease transmission and the lack of disease-related communication among the community groups. Although most participants recognized CL based on signs/symptoms, they lacked knowledge about its transmission. The lack of knowledge about CL transmission among the current study participants might be due to the absence of CL-targeted health education in the area. The present finding is consistent with most community-based KAP studies elsewhere, which reported better knowledge of CL symptoms but poor knowledge about its transmission [18, 23, 24, 29]. Community awareness about CL transmission might depend on the presence of health education in the area [30].

In this study, one-third of the respondents considered CL a stigmatizing disease. The observed stigma towards CL patients can be linked to the wrong perception of CL

transmission, seeing it as a contagious disease that might have existed among the present community groups. Depending on the extent of community perception and awareness of CL transmission, the stigma due to the disease can be higher or lower than the proportion presented in this study. A study in northwest Ethiopia reported a lower proportion (about one-fifth) of the respondents who perceived CL as a stigmatizing disease [23]. While in some areas, the proportion and the level of stigma could be higher than the present finding [31–36]. CL lesions and scars on the face are of considerable social impact due to stigma [11, 37–39].

About 77% of participants in this study responded that CL patients preferred to go to traditional healers rather than hospitals or clinics to receive treatment. This finding is similar to that of other studies among rural communities of Ethiopia: 68.3% in the Amhara region (northwest Ethiopia) [23], 90% in a district in the Tigray region (north Ethiopia) [18], and 67.6% in Gamo Gofa zone (south Ethiopia) [24]. In localities like the present study area, where modern CL treatment is unavailable, CL patients remain dependent on traditional therapies. The use of herbal preparation was a widely practiced traditional method to treat CL in the present study area, particularly in Kindo Didaye district, where most affected community groups live.

The very low prevention practices found in the current study, coupled with the absence of standard CL treatment, implies CL is a highly neglected disease in the area. These might have led most of the current study participants to consider CL a non-treatable disease. Community awareness about the treatability of CL might be better in areas where appropriate CL treatment is available and accessible [23, 40, 41]. However, the availability and accessibility of such treatment alone may not be sufficient to create community awareness toward CL treatability [24]. Engaging the community in health education programs would substantially improve the community's awareness of CL treatability.

In this study, the knowledge about CL was found to be associated with sex, age, and study districts. The male participants had better knowledge as compared to their female counterparts. In a male-dominated rural community [42], males might have better exposure to knowledge about CL. However, the association of knowledge about CL with gender was reported differently in some studies where females had better knowledge about the disease [18, 43]. In the current study, participants aged below or equal to 40 years had better knowledge. A relationship between CL knowledge and age was also shown in Tigray [18], although different age categories were used. In contrast, CL knowledge was better associated with the participants' age above 40 years in southwestern Yemen [29]. The differences in age-related CL knowledge in various areas might be due to the CL awareness of the communities [11, 29]. Interestingly, this study found that district was the most important factor associated with CL knowledge. Participants of the Kindo Didaye district had better knowledge than the other district. The observed district-wise difference in the knowledge about CL might be attributed to the eco-epidemiological factors and the associated experiences of the local communities between the districts regarding the disease [12, 40].

## Limitations

Our study was conducted among communities at higher altitudes. Hence, the findings of this study should not be generalized to the whole population of the zone. The face-to-face interview method of the data collection might have predisposed the respondents to social desirability bias. Nevertheless, we minimized the bias by restricting interviewers not to interview households from the same locality. Despite these limitations, this study presented valuable findings to emphasize health education campaigns and future CL prevention and control plans in the study areas.

## Conclusion

This study found a lack of knowledge about and practices toward CL in the study areas. The insufficient knowledge and community practice regarding the infection nature, vector, transmission ways, and treatment and presentation of CL emphasize the need for health education and community mobilization campaigns to increase community awareness about the disease in the area. The misconceptions observed about CL treatment also need the urgent establishment of treatment facilities.

## Declarations

### Consent for publication

Consent to publish this manuscript from the participants was deemed not applicable since the manuscript does not contain identifying data from any individual person.

## Supporting information

**S1 File. Questionnaire used for the data collection.**
(DOCX)

**S2 File. The dataset of KAP of CL, Wolaita zone, southern Ethiopia (August to October 2020).**
(SAV)

## Acknowledgments

We would like to thank VLIR-UOS, the University of Antwerp, Belgium, and Wolaita Sodo Universities for their technical and administrative support. We also extend our gratitude to the respective district administrative bodies, data collectors, data clerks, the local community, and study participants for their cooperation and technical assistance.

## Author Contributions

**Conceptualization:** Bereket Alemayehu, Herwig Leirs.

**Data curation:** Bereket Alemayehu, Mihiretu Alemayehu, Chimdesa Adugna.

**Formal analysis:** Bereket Alemayehu, Mihiretu Alemayehu.

**Funding acquisition:** Herwig Leirs.

**Investigation:** Bereket Alemayehu, Chimdesa Adugna, Tessema Bibo.

**Methodology:** Bereket Alemayehu, Abraham Getachew Kelbore, Mihiretu Alemayehu, Aberham Megaze, Herwig Leirs.

**Project administration:** Bereket Alemayehu, Aberham Megaze, Herwig Leirs.

**Resources:** Aberham Megaze, Herwig Leirs.

**Supervision:** Aberham Megaze, Herwig Leirs.

**Writing – original draft:** Bereket Alemayehu.

**Writing – review & editing:** Bereket Alemayehu, Abraham Getachew Kelbore, Mihiretu Alemayehu, Chimdesa Adugna, Tessema Bibo, Aberham Megaze, Herwig Leirs.

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
