## [Decision Letter · Decision Letter 0]

12 Jul 2022

PONE-D-22-15781Knowledge, attitude, and practice of the rural community about cutaneous leishmaniasis in Wolaita zone, southern EthiopiaPLOS ONE

Dear Dr. Bereket Alemayehu,

Thank you for submitting your manuscript to PLOS ONE. After careful consideration, we feel that it has merit but does not fully meet PLOS ONE’s publication criteria as it currently stands. Therefore, we invite you to submit a revised version of the manuscript that addresses the points raised during the review process.

We look forward to receiving your revised manuscript.

Kind regards,

Alireza Badirzadeh

Academic Editor

PLOS ONE

Journal Requirements:

We would like to thank VLIR-UOS, the University of Antwerp, Belgium, and Wolaita Sodo Universities for financial, technical, and administrative support. We also extend our gratitude to the respective district administrative bodies, data collectors, data clerks, local community, and study participants for their cooperation and technical assistance.

 This work was funded by the Flemish Interuniversity Council (VLIR-UOS, ET2019TEA485A102). The funders had no role in study design, data collection, and analysis, decision to publish, or preparation of the manuscript. 

5. We note that Figure 1 in your submission contain map/satellite images which may be copyrighted. All PLOS content is published under the Creative Commons Attribution License (CC BY 4.0), which means that the manuscript, images, and Supporting Information files will be freely available online, and any third party is permitted to access, download, copy, distribute, and use these materials in any way, even commercially, with proper attribution. For these reasons, we cannot publish previously copyrighted maps or satellite images created using proprietary data, such as Google software (Google Maps, Street View, and Earth). For more information, see our copyright guidelines: http://journals.plos.org/plosone/s/licenses-and-copyright.

a) You may seek permission from the original copyright holder of Figure 1 to publish the content specifically under the CC BY 4.0 license.  

Reviewers' comments:

Reviewer's Responses to Questions

**Comments to the Author**

1. Is the manuscript technically sound, and do the data support the conclusions?

Reviewer #1: Yes

Reviewer #2: Partly

Reviewer #3: Partly

2. Has the statistical analysis been performed appropriately and rigorously? 

Reviewer #1: Yes

Reviewer #2: No

Reviewer #3: Yes

3. Have the authors made all data underlying the findings in their manuscript fully available?

Reviewer #1: Yes

Reviewer #2: No

Reviewer #3: Yes

4. Is the manuscript presented in an intelligible fashion and written in standard English?

Reviewer #1: Yes

Reviewer #2: Yes

Reviewer #3: Yes

5. Review Comments to the Author

Reviewer #1: I reviewed a manuscript entitled “Knowledge, attitude, and practice of the rural community about cutaneous leishmaniasis in Wolaita zone, southern Ethiopia”. The work is intended to investigate the knowledge, attitude, and practice (KAP) of the community toward CL in the Wolaita zone, southern Ethiopia. The study concludes that the overall knowledge, attitude, and practice about CL and its prevention in the study area were low. The study is interesting and such types of studies provide the baseline information for disease control and management in endemic areas. The study has merit and should be considered for publication; however, I recommend minor revision before its publication in PLoS One.

Factors associated with knowledge about CL are among the key results of the study, however, these results are not mentioned in the abstract. I would suggest to please mention these results in the abstract.

In the discussion, the authors should also include some studies from other countries to support the current findings. This will add further diversity to the findings and will make the study of more international interest. Some examples of such studies are suggested below.

1. Doe ED, Egyir-Yawson A, Kwakye-Nuako G. Knowledge, attitude, and practices related to cutaneous leishmaniasis in endemic communities in the Volta region of Ghana. Int J Health Sci. 2019; 7(1):12.

2. Moussa S, Alshammari T, Alhudaires K, Alshammari T, Alshammari T, Elgendy A. Awareness, and behavioural practice of cutaneous leishmaniasis among hail population, Kingdom of Saudi Arabia. J Microbiol Exp. 2019; 7(2):88–9.

3. Ahmad S, Obaid MK, Taimur M, Shaheen H, Khan SN, Niaz S, Ali R, Haleem S. Knowledge, attitude, and practices towards cutaneous leishmaniasis in referral cases with cutaneous lesions: A cross-sectional survey in remote districts of southern Khyber Pakhtunkhwa, Pakistan. PLoS One. 2022 26;17(5): e0268801. doi: 10.1371/journal.pone.0268801. PMID: 35617283; PMCID: PMC9135282.

4. Zeinali M, Mohebali M, Mahmoudi M, Hassanpour GR, Shirza di MR. Study on knowledge, attitude, and practice of health workers of East Azerbaijan, Ilam and Khorasan Razavi provinces about leishmaniasis during 2015–2016: a comparative study before and after the intervention. Arch Clin Infect Dis. 2019; 14(1): e64282

The references should be revised as per the PLoS One author’s guidelines https://journals.plos.org/plosone/s/submission-guidelines. In some references the journal name is not abbreviated.

Reviewer #2: The article is relatively well written and is about CL in an area from which CL reports are scarce. However, I have a few issues with the article that should be improved before it can be considered for publication

MAJOR COMMENTS

The rationale for this study is not well explained in the introduction at all. The introduction is very general and does not explain what is already known and what is still unclear and why this is relevant. The introduction should be shortened, all the general stuff about CL in other countries can be removed, the different subspecies and the parts about eco-epidemiology are not related to this manuscript and are distracting. The introduction should be rewritten focused on why a KAP study is needed, what is already known and what is not yet known, also why a KAP would help in control. If there are already several KAP studies done in Ethiopia, it should be very clear why another one is needed. Simply repeating something in a new district is not enough to sell a study.

The sampling of the study subjects is not clear to me and should be elaborated or mentioned in the discussion as a limitation. Why were household heads chosen? Is this really the right population to study KAP? Is this a representative population for your community survey? Are they generally the patients who will be affected by CL or is CL more common in children and females in this area? The questions related to treatment practices may be better asked to patients who are actually suffering from CL rather than household heads not affected by CL.

The authors categorize all the outcome variables into Good and Bad, but I don’t really see the added value of this. What is the value of this classification? Is this a standardized tool that they were using? It would be useful to add the questionnaire as a supplementary file.

To me it would be more useful to simply describe the knowledge, attitude and practices rather than look at associated factors, especially because the sampling setup doesn’t lead to a representative population.

Missing data is not clearly described. Per variable, the number of missing values should be given. Additionally, the number of people who refused to participate should be mentioned.

What does it mean to have good knowledge of CL? Which variables are classified as ‘ correct’ and which are ‘ wrong’ ? My worry is that these things are not black and white, not all CL lesions are painful or disfiguring, how do you assess whether participants know a sandfly or not or whether they know CL or not? These things should be more clearly described.

How the attitude part of the questionnaire was done is unclear. It is mentioned that likert scales were done, ranging from 0 (strongly disagree) to 5 (strongly agree), but that the total attitude score ranged from 0-5. I would assume a maximum score of 25 would be possible if a patient strongly agreed with all statements? The attitude questions are classified as having unfavourable attitude and favourable attitude, but it is unclear what is really meant by this as the term is vague and seems to be more about treatment than the disease itself. Why is considered negative if a disease is not treatable or a health problem or not? I would suggest to simply describe attitudes and not classify them as good and bad.

Similarly, for practices I would advise to simply describe, rather than calling it good or bad practices. Why is it good or bad practice for CL prevention to live with domestic animals? Why is it poor practice to go to traditional healers?

In addition, I think asking about CL treatment and prevention practices is more appropriately asked to CL patients rather than community members.

The authors do a regression analysis to look at factors associated with CL knowledge. I would advise them to remove this, as it doesn’t really make sense. Especially since the population is not representative, the classification into good or poor is not strong, and the rationale of it is not clear.

The discussion is a bit long, and there is some repetition. I would advise the authors to make it more concise. Important things to add are a section on limitations and the relevance of this manuscript.

MINOR COMMENTS (also see PDF)

Were there any CL patients among the recruited participants?

It is not clear what the sample size is based on, is that knowledge? Attitude? Practices? This should be specified

Something more about the burden of CL in this area should be explained, what is the endemicity of the two districts and the villages that were sampled?

Classification of variables is not clearly explained. Why was age cutoff of 40 used? How was no formal education classified? Why was a family size of 5 used as a cutoff?

Reviewer #3: Thank you for the opportunity to review the manuscript titled “Knowledge, attitude, and practice of the rural community about cutaneous leishmaniasis in Wolaita zone, southern Ethiopia”. I would like to appreciate the authors' effort to undertake this valuable task. Based on my point of view I raised the following points that I believed might improve the manuscript. This work plans to assess the community's knowledge, attitude, and practice about cutaneous leishmaniasis in Wolaita zone, southern Ethiopia. This kind of researches provides baseline information for future studies. However, there are some concerns which when addressed hope to improve the quality of the manuscript. Find my specific comments below.

1- Line 42, “What plants were used in herbal treatment?

2- Line 54, “Cutaneous leishmaniasis…. For the first time, write it completely and then use its abbreviation (CL)

3- Line 63, “In 2020, about 85% of new….. The reference related to this sentence is for 2010.

4- Line 65, East African countries, including Ethiopia, are…… There is a duplicate with the last paragraph of the introduction.

5- Table 1: Instead of level of education I would recommend use of “years of education”. Reads might not be familiar with Ethiopian system of education.

6- Did you have any missing data among the variables?

7- I suggest the authors put raw data in an online open access repository for preserve and share their research outputs.

8- Line 251, “ perhaps??? In the south of Ethiopia, there are similar studies, please write clearly whether there is a similar study in your studied area.

References

9- Italicize the scientific names (Leishmania tropica in REF 13)

10- Journal names should appear in abbreviation.

11- No journal name should start with "the"

12- Books need city.

6. PLOS authors have the option to publish the peer review history of their article (what does this mean?). If published, this will include your full peer review and any attached files.

Reviewer #1: **Yes: **Rehman Ali

Reviewer #2: No

Reviewer #3: No

---

## [Author Response · Author response to Decision Letter 0]

13 Oct 2022

All comments of the reviewers and the editor are addressed and attached in a separate file (Response to Reviewers).

---

## [Decision Letter · Decision Letter 1]

21 Nov 2022

PONE-D-22-15781R1Knowledge, attitude, and practice of the rural community about cutaneous leishmaniasis in Wolaita zone, southern EthiopiaPLOS ONE

Dear Dr. Bereket Alemayehu,

Thank you for submitting your manuscript to PLOS ONE. After careful consideration, we feel that it has merit but does not fully meet PLOS ONE’s publication criteria as it currently stands. Therefore, we invite you to submit a revised version of the manuscript that addresses the points raised during the review process.

We look forward to receiving your revised manuscript.

Kind regards,

Alireza Badirzadeh

Academic Editor

PLOS ONE

Journal Requirements:

Reviewers' comments:

Reviewer's Responses to Questions

**Comments to the Author**

1. If the authors have adequately addressed your comments raised in a previous round of review and you feel that this manuscript is now acceptable for publication, you may indicate that here to bypass the “Comments to the Author” section, enter your conflict of interest statement in the “Confidential to Editor” section, and submit your "Accept" recommendation.

Reviewer #1: All comments have been addressed

Reviewer #2: (No Response)

2. Is the manuscript technically sound, and do the data support the conclusions?

Reviewer #1: Yes

Reviewer #2: Yes

3. Has the statistical analysis been performed appropriately and rigorously? 

Reviewer #1: Yes

Reviewer #2: Yes

4. Have the authors made all data underlying the findings in their manuscript fully available?

Reviewer #1: Yes

Reviewer #2: No

5. Is the manuscript presented in an intelligible fashion and written in standard English?

Reviewer #1: Yes

Reviewer #2: Yes

6. Review Comments to the Author

Reviewer #1: All the comments have been addressed by the authors and I recommend the manuscript for publication in PLOS One.

Reviewer #2: Most of the comments have been adequately adressed.

I still have two main points that I am not satisfied with

-I still really disagree with categorizing attitudes as positive or negative. Firstly, I don't what see it adds to classify it overall, and secondly, the overall category (negative/positive attitude towards CL) doesn’t fit the underlying questions.

To me, classifying someone as having a positive or negative attitude is possible (e.g. the last two questions). But the questions about CL being treatable, CL patients having to receive modern medication and CL being a health problem don’t fit the description as good/bad attitude. Why is it negative attitude if someone considers CL a health problem?

How can you classify whether people CL as a health problem as positive or negative? And how can you classify CL is treatable

-The discussion is still very long and contains sections which are more suited for results.

Try to really link it to things that can be used for policy and practice and how the findings can be use to improve community engagement. I have made suggestions in the attached PDF.

7. PLOS authors have the option to publish the peer review history of their article (what does this mean?). If published, this will include your full peer review and any attached files.

Reviewer #1: No

Reviewer #2: No

---

## [Author Response · Author response to Decision Letter 1]

19 Jan 2023

The authors of this manuscript are thankful to the reviewers for the provided comments an suggestions. The point-by-point response letter is uploaded in a separate file in this submission.

---

## [Decision Letter · Decision Letter 2]

27 Feb 2023

PONE-D-22-15781R2Knowledge, attitude, and practice of the rural community about cutaneous leishmaniasis in Wolaita zone, southern EthiopiaPLOS ONE

Dear Dr. Bereket Alemayehu,

Thank you for submitting your manuscript to PLOS ONE. After careful consideration, we feel that it has merit but does not fully meet PLOS ONE’s publication criteria as it currently stands. Therefore, we invite you to submit a revised version of the manuscript that addresses the points raised during the review process.

ACADEMIC EDITOR:Please apply reviewer's comments.

We look forward to receiving your revised manuscript.

Kind regards,

Alireza Badirzadeh

Academic Editor

PLOS ONE

Journal Requirements:

Reviewers' comments:

Reviewer's Responses to Questions

**Comments to the Author**

1. If the authors have adequately addressed your comments raised in a previous round of review and you feel that this manuscript is now acceptable for publication, you may indicate that here to bypass the “Comments to the Author” section, enter your conflict of interest statement in the “Confidential to Editor” section, and submit your "Accept" recommendation.

Reviewer #1: (No Response)

Reviewer #2: (No Response)

2. Is the manuscript technically sound, and do the data support the conclusions?

Reviewer #1: Yes

Reviewer #2: Yes

3. Has the statistical analysis been performed appropriately and rigorously? 

Reviewer #1: Yes

Reviewer #2: Yes

4. Have the authors made all data underlying the findings in their manuscript fully available?

Reviewer #1: Yes

Reviewer #2: No

5. Is the manuscript presented in an intelligible fashion and written in standard English?

Reviewer #1: Yes

Reviewer #2: Yes

6. Review Comments to the Author

Reviewer #1: The raised concerns have been thoroughly addressed by the authors and I recommend the manuscript for publication in PLOS One

Reviewer #2: Thank you for your revised manuscript. I have no major comments left. Here I provide a few remaining suggestions to improve readability:

• Line 205-207: Here for binary responses both percentages are given (the n,% for those who had bad feelings and n,% for those who didn’t). This can be reduced as follows.

A significant number (185,205 43.8%) of the respondents had bad feelings about meeting CL patients. Most (70.9 %) respondents believed CL was not a stigmatizing disease (Table 3).

• Line 260: only one third thought CL to be stigmatizing. This seems low. It would be interesting to explain potential reasons for this, rather than stating things that can lead to stigma.

• Line 273: there seems to be a typo, should be Kindo Didaye (not Kndo)

• Line 289: "Studies also reported different findings concerning CL knowledge and age-wise association [18, 29]." It would be better to specify if the findings confirm your finding or contrast it.

o Ref 18 shows a relationship with age, but doesn’t show a difference in those below and above 40

o Ref 29 doesn’t show a relationship with age.

o Suggestion to rewrite: A relationship between CL knowledge and age was also shown in Tigray [18], although different age categories were used.

• Line 300: better to replace misinterpreted with 'generalized'

• Line 311: campaign(s), should be plural

• Some references should be checked

o Ref 5 WHOECotCot?

o Ref 15: initials should come after the last name

o Ref 28: WHO is not an author here

7. PLOS authors have the option to publish the peer review history of their article (what does this mean?). If published, this will include your full peer review and any attached files.

Reviewer #1: No

Reviewer #2: No

---

## [Author Response · Author response to Decision Letter 2]

2 Mar 2023

Response to Reviewers (PLOS ONE)

Manuscript Number: PONE-D-22-15781R2

Knowledge, attitude, and practice of the rural community about cutaneous leishmaniasis in Wolaita zone, southern Ethiopia

We thank the editor and the reviewers for giving us feed-backs on the manuscript. We also appreciate PLoS One/editor for allowing us to clean our manuscript further. Please find below our responses to the points raised in the current review. In this response letter, the editor’s and the reviewer’s comments/suggestions are in blue, and our responses are in black. We have also made the requested changes in the manuscript (yellow colored). We hope the recent revisions now address the comments the academic editor and reviewer pointed out.

Academic Editor's Comments:

Journal Requirements:

Comment #1. Please review your reference list to ensure that it is complete and correct. If you have cited papers that have been retracted, please include the rationale for doing so in the manuscript text, or remove these references and replace them with relevant current references. Any changes to the reference list should be mentioned in the rebuttal letter that accompanies your revised manuscript. If you need to cite a retracted article, indicate the article’s retracted status in the References list and also include a citation and full reference for the retraction notice.

Response: Thank you for the concern about the completeness and correctness of references. We ensure that all references are complete and correct and all are relevant. 

Reviewer’s Comments:

Reviewer #2: Thank you for your revised manuscript. I have no major comments left. Here I provide a few remaining suggestions to improve readability:

• Line 205-207: Here for binary responses both percentages are given (the n,% for those who had bad feelings and n,% for those who didn’t). This can be reduced as follows.

A significant number (185,205 43.8%) of the respondents had bad feelings about meeting CL patients. Most (70.9 %) respondents believed CL was not a stigmatizing disease (Table 3).

Response: We thank the reviewer for the comments and suggestions. We have accepted the comments/suggestions (lines 206 to 208 of the revised manuscript).

• Line 260: only one third thought CL to be stigmatizing. This seems low. It would be interesting to explain potential reasons for this, rather than stating things that can lead to stigma.

Response: We appreciate the reviewer for this suggestion. We revised the section (lines 259 to 263). 

• Line 273: there seems to be a typo, should be Kindo Didaye (not Kndo)

Response: Thank you. The typo is corrected (line 273).

• Line 289: "Studies also reported different findings concerning CL knowledge and age-wise association [18, 29]." It would be better to specify if the findings confirm your finding or contrast it.

Response: We accepted the comment. Comparisons to the findings are specified (lines 289 to 293). 

o Ref 18 shows a relationship with age, but doesn’t show a difference in those below and above 40

Response: We revised it now (lines 289/290). 

o Ref 29 doesn’t show a relationship with age.

Response: Ref 29 presents the CL knowledge and age-wise association (on page 9, Table 6 of the reference paper). In the reference paper, age >40 was associated with CL knowledge, whch contrasts our finding (lines 291/292). 

o Suggestion to rewrite: A relationship between CL knowledge and age was also shown in Tigray [18], although different age categories were used.

Response: Thanks. We accepted the suggestion (lines 289/290). 

• Line 300: better to replace misinterpreted with 'generalized'

Response: We accepted the suggestion (line 303).

• Line 311: campaign(s), should be plural

Response: We accepted the comment (line 313).

• Some references should be checked

o Ref 5 WHOECotCot?

o Ref 15: initials should come after the last name

o Ref 28: WHO is not an author here

Response: Thanks. References are checked. We revised the section.

---

## [Decision Letter · Decision Letter 3]

14 Mar 2023

Knowledge, attitude, and practice of the rural community about cutaneous leishmaniasis in Wolaita zone, southern Ethiopia

PONE-D-22-15781R3

Dear Dr. Bereket Alemayehu,

We’re pleased to inform you that your manuscript has been judged scientifically suitable for publication and will be formally accepted for publication once it meets all outstanding technical requirements.

Kind regards,

Alireza Badirzadeh

Academic Editor

PLOS ONE

Additional Editor Comments (optional):

Reviewers' comments:

Reviewer's Responses to Questions

**Comments to the Author**

1. If the authors have adequately addressed your comments raised in a previous round of review and you feel that this manuscript is now acceptable for publication, you may indicate that here to bypass the “Comments to the Author” section, enter your conflict of interest statement in the “Confidential to Editor” section, and submit your "Accept" recommendation.

Reviewer #2: All comments have been addressed

2. Is the manuscript technically sound, and do the data support the conclusions?

Reviewer #2: Yes

3. Has the statistical analysis been performed appropriately and rigorously? 

Reviewer #2: Yes

4. Have the authors made all data underlying the findings in their manuscript fully available?

Reviewer #2: No

5. Is the manuscript presented in an intelligible fashion and written in standard English?

Reviewer #2: Yes

6. Review Comments to the Author

Reviewer #2: (No Response)

7. PLOS authors have the option to publish the peer review history of their article (what does this mean?). If published, this will include your full peer review and any attached files.

Reviewer #2: No

---

## [Editor Report · Acceptance letter]

17 Mar 2023

PONE-D-22-15781R3 

Knowledge, attitude, and practice of the rural community about cutaneous leishmaniasis in Wolaita zone, southern Ethiopia 

Dear Dr. Alemayehu:

I'm pleased to inform you that your manuscript has been deemed suitable for publication in PLOS ONE. Congratulations! Your manuscript is now with our production department. 

Kind regards, 

on behalf of

Dr. Alireza Badirzadeh 

Academic Editor

PLOS ONE